# Proinflammatory Dietary Intake is Associated with Increased Risk of Metabolic Syndrome and Its Components: Results from the Population-Based Prospective Study

**DOI:** 10.3390/nu12041196

**Published:** 2020-04-24

**Authors:** Imran Khan, Minji Kwon, Nitin Shivappa, James R. Hébert, Mi Kyung Kim

**Affiliations:** 1Cancer Epidemiology Branch, Division of Cancer Epidemiology and Prevention, National Cancer Center, Goyang 10408, Gyeonggi-do, Korea; imrankhan572@yahoo.com (I.K.); kkmj2020@gmail.com (M.K.); 2Cancer Prevention and Control Program, University of South Carolina, Columbia, SC 29208, USA; shivappa@email.sc.edu (N.S.); JHEBERT@mailbox.sc.edu (J.R.H.); 3Department of Epidemiology and Biostatistics, Arnold School of Public Health, University of South Carolina, Columbia, SC 29208, USA; 4Department of Nutrition, Connecting Health Innovations LLC, Columbia, SC 29201, USA

**Keywords:** chronic inflammation, dietary inflammatory index, HDL-C, KoGES cohort, metabolic syndrome

## Abstract

Metabolic syndrome (MetS) is a major public health challenge throughout the world, although studies on its association with the inflammatory potential of diet are inconsistent. The aim of this prospective study was to assess the association between the Dietary Inflammatory Index (DII^®^) and the risk of MetS and its components in a Korean population. Data from 157,812 Korean adults (mean age 52.8 years; 53,304 men and 104,508 women with mean follow-up of 7.4 years) collected by members of the Korean Genome and Epidemiology Study form the basis for this report. DII scores were calculated based on Semi-Quantitative Food-Frequency Questionnaire data. Multivariable-adjusted Cox proportional hazard models were used to estimate the association between DII scores and MetS. In women, higher DII scores (pro-inflammatory diet) increased the risk of MetS (hazard ratio [HR]_quintile5 v. 1_ 1.43; 95% confidence interval (CI) 1.21–1.69; *p* for trend ≤ 0.0001) and its five components. A positive association was observed for postmenopausal women, with a 50% higher risk of developing MetS (HR_quintile5 v. 1_ 1.50; 95% CI 1.23–1.83; *p* for trend = 0.0008) after fully adjusting for potential confounders. Irrespective of the menopausal status of women, higher DII (=Q5) scores were positively associated with all 5 components of MetS (*p* < 0.05). In men, higher DII scores significantly increased the risk of low HDL cholesterol [HR]_quintile5 v. 1_ 1.59 (1.27–1.99); *p* for trend = 0.0001], elevated waist circumferences [HR]_quintile5 v. 1_ 1.28 (1.08–1.52); *p* for trend = 0.01], and high blood pressure [HR]_quintile5 v. 1_ 1.17 (1.03–1.32); *p* for trend = 0.05]. These results indicate that diet with pro-inflammatory potential, as represented by higher DII scores, is prospectively associated with increased risk of MetS, and the relationship is stronger in women than in men.

## 1. Introduction

Metabolic syndrome (MetS) is a multifactorial disorder characterized by elevated fasting glucose, triglycerides, blood pressure, and waist circumference and low levels of high-density lipoprotein cholesterol (HDL-C) levels [1]. MetS, as well as its components, have become a major public health challenge around the world, as they are known risk factors for cardiovascular diseases (CVD) and type 2 diabetes mellitus [2,3]. MetS has been a scourge globally, including in both developed and developing countries [4]. In the US, it is estimated that nearly 35% of all adults and 50% of those of aged 60 years or older have MetS in 2011–12 [5]. The overall age-standardized prevalence of MetS was 22.4% in 2015, and nearly 40% of women aged 60 years or older in South Korea had MetS in 2013–2015 [6]. 

Chronic inflammation, characterized by continuous presence of chronic, low-grade systemic inflammation plays a major role in the pathophysiology of many chronic diseases, including MetS [7]. Continuous exposure to chronic stressors (e.g., tobacco smoking, chronic infection, obesity) may contribute to chronic inflammation, which leads to the onset of many chronic diseases over time, resulting in MetS [3]. Several risk factors, including sex, age, smoking, physical activity, and several dietary components are known to contribute to the progression of this chronic inflammatory condition [8]. Diet plays an important role in the regulation of the inflammatory process as it affects the balance between anti- and pro-inflammatory cytokines and adipokines [7,8,9]. Diets rich in vegetables, fruits, antioxidants, vitamins and healthy oils decrease chronic systemic inflammation, while those rich in fats and simple carbohydrates increase inflammation [3]. The Dietary Inflammatory Index (DII^®^) was designed to comprehensively measure the inflammatory effect of an individual’s diet [10]. Numerous studies have established a positive association between DII scores and blood inflammatory biomarkers (e.g., high-sensitivity C-reactive protein [CRP], homocysteine, interleukin-6 and tumor necrosis factor) [11,12,13,14].

The DII has been shown to be associated with a number of health outcomes including cancer, CVD, adverse mental health and musculoskeletal disorders [15,16,17,18,19]. However, the association between DII and MetS has been inconsistent across studies. No association was reported in previous cross-sectional [20,21] and cohort studies [22], although individual components of MetS have been associated with the DII in cross-sectional studies for which an overall association with MetS was not found [20,21]. One cohort study performed in France, with mean follow up of 12.4 years, revealed a positive association between DII scores and increased incidence of MetS [23]. So far, only two studies on the DII and MetS have been reported from Asia, and both reported a positive association between DII and MetS [24,25]. However, both of these studies are cross-sectional in design, have small population sizes and obtained data on a limited food parameters for the calculation of DII (23 items for the Korean study and 30 items for the Iranian study); thus, making it hard to generalize the results to the whole population or to other populations. With this as background, we aimed to investigate the association between DII and MetS and its components in a large Korean population-based prospective study. 

## 2. Materials and Methods

### 2.1. Cohort Characteristics 

Korean Genome and Epidemiological Studies Health Examination cohort (KoGES_HEXA) data were used for the present study; detailed information for which can be found elsewhere [26]. Briefly, both men and women participants aged 40–79 years were recruited by the National Health Examinee Registry at baseline for this study. This cohort enrolled new participants mostly from examination/medical institutions all over the country. Annually, about 30,000 people in 38 health examination centers and hospitals participate in the survey. Different means such as on-site invitation, letters, campaign and community conferences have been used to recruit participants who voluntarily fill out a baseline survey. A consent form was signed by all the participants at the beginning of the study. The Institutional Review Board of the National Cancer Center and the institutions that had participated in the KoGES cohort approved the study design (IRB No. NCC2018-0164). Between 2004 and 2013 a total of 173,343 participants were enrolled, and these subjects now have a mean follow-up time of 7.4 years. Data generated at baseline to the initial follow-up were utilized for this report. Participants who had MetS at the baseline (n =11,257) or had missing dietary information (n = 4278) were excluded from these analyses. MetS was diagnosed among the participants based on the Modified National Cholesterol Education Program Adult Treatment Panel III (NCEP ATP III) and the obesity guidelines of the Korean Society for the Study of Obesity (KSSO), as ≥3 of any of the following [27,28]: waist circumference [WC] (≥90 cm for men or ≥85 cm for women); high triglyceride level (≥150 mg/dL), low HDL-C level (<40 mg/dL in men or <50 mg/dL in women); high glucose level (fasting plasma glucose level ≥100 mg/dL) and high blood pressure (systolic blood pressure/diastolic blood pressure ≥130/85 mm Hg or the use of antihypertensive drugs). A total of 3507 cases (men = 1395; women = 2112) of MetS were identified (Figure 1).

### 2.2. Computation of the Dietary Inflammatory Index (DII^®^)

Dietary assessment of participants at baseline was evaluated using a semi-quantitative food frequency questionnaire (SQ-FFQ). Comprehensive information on the validity of the SQ-FFQ is available elsewhere [29,30]. Based on the SQ-FFQ, a total 106 food items were included and the participants were asked to assess their consumption frequencies and the average amounts consumed for the whole year. To calculate the nutrient consumption per day, the total of the values of the average serving amounts, portions per unit, and serving frequencies was applied [31]. The frequencies of each food item were calculated according to nine choices, starting from “almost never” to “more than three per day”, while portion size was estimated from three response choices such as ½ serving, 1 serving, and 1.5 servings and daily nutrient intakes were estimated using a food composition table [32]. 

Details of the DII are available elsewhere [10]. Briefly, a total of 1943 research articles were reviewed and scored for 45 food parameters based on their effects on the levels of inflammatory markers such as CRP, IL-4, IL-6, IL-1β, IL-10, and TNF-α. In the current study a total of 37 parameters out of 45 were available. The nutritional data used in the current study were measured from the Functional Ingredients Table (Rural Development Administration), Computer Aided Nutritional Analysis (Korean Nutrition Society), and the U.S. Department of Agriculture. Global databases of diet surveys from 11 countries, including Korea, for each of the 45 parameters (i.e., foods, nutrients, and other food constituents) were used to calculate DII score [10]. As explained in a recent article delineating improvements in this latest version of the scoring algorithm, higher DII scores indicate more pro-inflammatory diets, while the lower DII scores indicate more anti-inflammatory diets [33].

### 2.3. Covariates

The information about sociodemographic characteristics, personal and family history, physical activity and SQ-FFQ were obtained at the baseline and follow-up medical examinations. Both continuous and categorical variable were assessed and the details can be found elsewhere [34].

### 2.4. Statistical Analysis

For all statistical analysis, the SAS^®^ 9.3 (SAS Institute, Cary, NC, USA) program was used. The participants were divided into five groups (quintiles) based on their DII scores. DII quintiles were obtained from the cohort with no MetS diagnosis at baseline. Continuous variables were represented as means with standard deviation, while categorical variables were represented as frequency numbers with percentage, respectively. The Jonckheere–Terpstra test and Mantel–Haenszel Chi-square test was used to calculate *p* for trend values for continuous and categorical variables, respectively. The Kruskal–Wallis test was used to identify significant difference between the normal and MetS groups for distribution of food parameters and nutrients of DII. The association between the baseline DII and MetS were analyzed through multivariate Cox proportional hazard model. To confirm the assumption of proportional risk, all of the models were evaluated and deemed to be consistent with a model that included time-dependent covariates. The model was adjusted for these variables: sex, age, smoke, alcohol drinking, physical activity, body mass index (BMI), family history of diabetes mellitus, family history of hypertension and energy intake. Stratified analyses were carried out by smoking and menopausal status in women. The hazard ratios (HRs) were calculated along with 95% confidence intervals (CIs); and two-sided probability values <0.05 were considered statistically significant. We assessed the heterogeneity between sex, menopausal status and smoking status in subgroups, using a likelihood ratio test comparing Cox proportional hazard models with and without interaction terms for DII and subgroups. Throughout, *p* values < 0.05 were considered statistically significant.

## 3. Results 

The demographic characteristics of the study participants at the baseline based on the DII quintiles are presented in Table 1. A total of 157,812 participants (male = 53,304; female = 104,508) with evaluable data were included for final analysis. The range of scores in the highest DII quintile is 2.19 to 6.93, while scores in the lowest DII quintile range from −9.12 to −0.97. The median DII value for men was higher (0.94) than women (0.89). The mean age of the participants was 52.9 ± 8.3 years (53.9 ± 8.8 in men and 52.4 ± 8.0 in women). The DII scores were increased as the mean age of the participants was increased according to the quintiles (*p* < 0.0001). By contrast, the DII was decreased as the educational attainment and income of the participants were higher according to the quintiles (*p* < 0.0001). The percentage of current smokers increased and never smokers decreased by DII quintile, while the numbers of current drinkers decreased and never drinkers increased as the DII increased (*p* < 0.0001). Participants with higher DII score were physically inactive as compared to those with lower score (*p* < 0.0001). The percentage of married people decreased and post-menopausal women increased as the DII increased (*p* < 0.0001). The mean value of waist circumference, triglyceride levels, fasting glucose levels and systolic and diastolic blood pressure were higher in men than in women. However, mean HDL cholesterol levels were higher in women than in men. 

A total of 3507 individuals (1395 men and 2112 women) were identified with MetS, during the 7.4 years’ follow-up. The incidence of MetS was higher for women (1.34%) than men (0.88%). Multivariate Cox proportional hazard analysis (Table 2) showed a more pro-inflammatory diet (=Q5) was found to have a greater risk of developing MetS (HR _quintile5 v. 1_ 1.31; 95% CI 1.15–1.49; *p* for trend = 0.002) or individual components of MetS such as WC (HR _quintile5 v. 1_ 1.37; 95% CI 1.24–1.51; *p* for trend ≤ 0.0001), high triglycerides (HR _quintile5 v. 1_ 1.24; 95% CI 1.14–1.35; *p* for trend ≤ 0.0001), low HDL-C (HR _quintile5 v. 1_ 1.63; 95% CI 1.44–1.84; *p* for trend ≤ 0.0001), hyperglycemia (HR _quintile5 v. 1_ 1.18; 95% CI 1.09–1.26; *p* for trend ≤ 0.0001) and high blood pressure (HR _quintile5 v. 1_ 1.24; 95% CI 1.15–1.34; *p* for trend ≤ 0.0001) after adjusting for potential confounding variables such as sex, age, smoke, alcohol drinking, physical activity, BMI, family history of diabetes mellitus, family history of hypertension and energy intake. After stratification by sex, women were found to have a 43% higher risk of developing MetS (HR _quintile5 v. 1_ 1.43; 95% CI 1.21–1.69; *p* for trend ≤ 0.0001) and its 5 components, while waist circumference (HR _quintile5 v. 1_ 1.28; 95% CI 1.08–1.52; *p* for trend = 0.01), HDL-C (HR _quintile5 v. 1_ 1.59; 95% CI 1.27–1.99; *p* for trend = 0.0001), and blood pressure (HR _quintile5 v. 1_ 1.17; 95% CI 1.03–1.32; *p* for trend = 0.05) were significantly associated with higher DII scores in men.

Table 3 shows the association of DII scores with the MetS and its components in women after stratification by menopausal status. A positive association was observed for postmenopausal women, with a 50% higher risk of developing MetS (HR _quintile5 v. 1_ 1.50; 95% CI 1.23–1.83; *p* for trend = 0.0008) after being fully adjusted for potential confounders. Irrespective of the menopausal status of women, higher DII (=Q5) were positively associated with 5 components of MetS (*p* < 0.05).

After stratifying by smoking status, it was revealed that smokers were more likely to have a higher risk of MetS-components, as compared to non-smokers (Table 4). The association was stronger for elevated waist circumferences (HR _quintile5 v. 1_ 2.30; 95% CI 1.04-5.06), low HDL-C (HR _quintile5 v. 1_ 2.66; 95% CI 1.20–5.90), and blood pressure (HR _quintile5 v. 1_ 2.71; 95% CI 1.41–5.18) among smoking women, as compared to non-smoking women. In men, the risk of elevated waist circumferences and low HDL-C were significantly elevated among smokers, compared to non-smokers. 

The distribution of the food and nutrient parameters of the DII^®^ among the 3507 MetS cases and 154,305 controls are presented in Appendix A. Statistically significant differences were found for anti-inflammatory foods and nutrients such as flavones, flavonones, onion, magnesium, selenium and vitamins B1, B2, C, E, and folic acid among the controls and cases. Significant differences were observed for pro-inflammatory dietary components such as energy, total fat, saturated fat, cholesterol, carbohydrate, and iron between cases and controls. The odds ratios of metabolic syndrome for quintiles of anti-inflammatory food and nutrient parameters are presented in Appendix A. Significantly higher intakes of flavan-3ol, flavonols, tea, iso-flavones, PUFA, onion, fiber, folic acid, n-3 PUFA, protein, vitamins A, B1, B2, B6, C, and E intakes were associated with the risk of metabolic syndrome. 

## 4. Discussion

In the present study we investigated the association between DII score and MetS in a representative Korean population. We found that higher DII score was significantly associated with higher risk of MetS and its components only in women participants after adjusting for potential confounders. Higher DII scores were positively associated with MetS in postmenopausal women. 

Various studies have investigated the association between DII and MetS, however, the results have been inconsistent. For example, a prospective study conducted in a Spanish cohort of university graduates (n = 6851) with a median follow-up of 8.3 years examined the association between different dietary indexes and MetS incidence [21]. No significant association between the DII and MetS was observed. Other cross-sectional studies also found no significant association between the DII and MetS, including the Buffalo Cardio-Metabolic Occupational Police Stress (BCOPS) study conducted in the USA (n = 464), Polish-Norwegian (PONS) study (n = 3862), Luxembourg study (n = 1352), and Lebanese study (n = 331) [20,21,35,36]. Obviously, all of these studies have much smaller population sizes than the current study.

In line with our results, the prospective Supplementation en Vitamines et Mineraux AntioXydants (SUVIMAX) cohort study (n = 3726), conducted in France, investigated the association between the DII and MetS. With an average follow-up of 12.4 years they found an increased risk of developing MetS with the highest DII scores (OR 1.39; 95% CI 1.01–1.92) [23]. Similarly, a cross-sectional study performed in Iran, including a total of 606 participants from East-Azarbaijan-Iran, showed a significant association between the higher DII score and MetS (OR 2.26; 95% CI 1.03–4.92), after adjusting for various confounders [25]. Contrary to our findings that higher DII scores were significantly associated with MetS in women only, a cross-sectional study conducted in Korea, which used Korea National Health and Nutrition Examination Survey (KNHANES) data (2013–2015; n = 9291), found that higher DII scores were significantly associated with MetS prevalence only in men (OR 1.40; *p* for trend = 0.008) [25]. Although in postmenopausal women, the positive association between DII and MetS (OR 1.67, 95% CI 1.15–2.44; *p* for trend = 0.008) was similar to our findings. 

The inconsistency in the results from other studies could be attributed to many factors, such as differences in study design, dietary characteristics, sample size, region, and age of subjects [20,21,22,23,24,35,36,37]. For example, the SUVIMAX study [23] had used 24-h dietary records to obtain 36 food parameters, including anti-inflammatory foods and flavonoids; on the other hand, some studies reported only 22–28 food parameters and without anti-inflammatory spice parameters [20,21,22,35,36,37]. By contrast, in the current study we have used 37 food parameters from the FFQ and these included onion, garlic, and tea (black and green) as anti-inflammatory foods, as well as nutrient parameters to calculate DII. 

The reasons for a relation between MetS and inflammation are not fully elucidated. However, one possible reason could be that adipose tissue in obese individuals with MetS releases a higher amount of cytokines in the blood stream [38]; this, in turn, accounts for a greater production of CRP by the liver [39]. The production of pro-inflammatory mediators could be triggered by one’s individual genetic makeup. Further to this, it has been suggested that the effect of diet on inflammation is dependent on genetics; i.e., polymorphism in genes responsible for the production of cytokines [40]. Another possible reason is that insulin resistance *per se* is responsible for a higher production of cytokines [41]. 

The DII was significantly associated with all the five components of MetS in women, while only waist circumference and HDL-C were associated in men. Again for MetS-components, the results are inconsistent in various studies. In SUVIMAX study [23], high DII scores were associated with increased blood pressure and triglyceride levels and low HDL-C levels, while in the Iranian Lifestyle Promotion Project study [25], DII was significantly associated with increased fasting blood glucose level. Instead, no association between DII and MetS-components was observed in the Lebanese and Luxembourg studies [35,36]. These findings suggest that the association between DII and MetS-components may vary across populations, and future work is warranted to draw a clear conclusion. 

Previously, it was reported that dietary inflammatory potential of diet seems to have a bigger effect on postmenopausal women who are susceptible to developing MetS [24]. In that Korean study, they found a positive association between the DII and MetS in postmenopausal women. This increased impact can be explained by the fact that the women after menopause lack estrogen hormone, which help to alter fat metabolism and, as a result, this promote MetS [42]. 

A growing body of evidence suggests that cigarette smoking can cause the onset of MetS, dyslipidemia, obesity, elevated blood pressure and blood glucose prior to CVDs [43,44]. However, in the present study, we did not find a significant association between DII and MetS in smoking men and women, despite an 88% higher risk in women. Although the MetS-components such as abnormal HDL-C levels and blood pressure were positively associated with higher DII scores in smoking women, in smoking men the DII was associated only with abnormal HDL-C levels.

In the present study, we found sex differences in the effect of DII on MetS and its components. Our findings are in line with previous studies, which found different association by sex [21,24]. The mechanism(s) behind the differential effect of DII on MetS-components by sex are not fully understood. However, a possible explanation is that body fat, especially abdominal fat and production of sex hormones, may contributed to the modulation of inflammatory markers and influence MetS-components [45,46,47]. 

The strengths of the current study include its prospective study design with a long follow-up of about 7.4 years, a large sample size and including participants free of MetS at the baseline. Another strength is the use of DII in the study, which is based on peer-reviewed literature focusing specifically on the effect of various dietary components and inflammation and the availability of 37 out of 45 food parameters to calculate DII. In addition, the current study findings may be of public health significance as a result of shifting from pro- to anti-inflammatory dietary approaches to protect CVDs. Despite its strengths, this study has some limitations. First, we used data from a single 24-h recall, which may not represent the usual intake of the subject. Second, we did not have information on specific higher triglycerides levels and HDL-C medications taken by the participants to include in the MetS criteria. Third, any change by the participants in diet after being diagnosed with MetS have not been captured, which could create an artifact of more pro-inflammatory DII scores not being associated, or even being protective, in some cases of MetS or its components. Finally, the associations between the DII and MetS found in this study may not be generalized to populations having very different dietary habits.

## 5. Conclusions

The inflammatory potential of the diet, as shown by higher DII scores, is prospectively associated with a higher risk of metabolic syndrome in women. These findings suggest a potential role of inflammation as an important underlying reason linking MetS and diet. 

## Figures and Tables

**Figure 1 nutrients-12-01196-f001:**
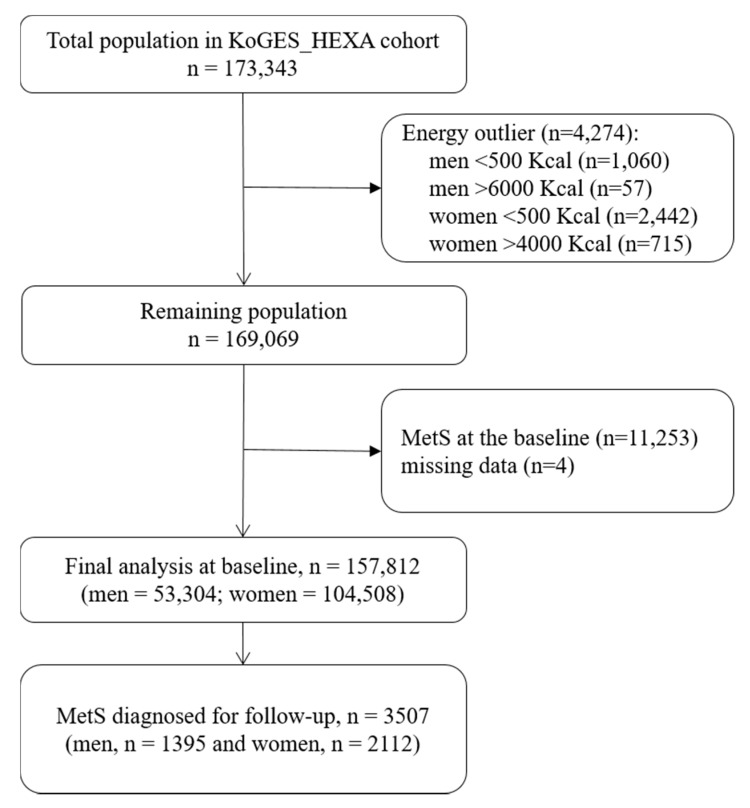
Flowchart of Korean Genome and Epidemiological Studies Health Examination (KoGES_HEXA) cohort participants included in the main analysis. MetS: Metabolic Syndrome.

**Table 1 nutrients-12-01196-t001:** Baseline characteristics of the study participants by Dietary Inflammatory Index^®^ (DII^®^) quintiles in the KoGES_HEXA cohort, 2004–2013.

Characteristics	Quintiles of Dietary Inflammatory Index^® a^					*p* for Trend ^b^
	Q1	Q2	Q3	Q4	Q5	
Total observations (n = 157,812)	31,542	31,565	31,593	31,555	31,557	
Total cases of MS (n = 3507)	681	704	732	694	696	
DII^®^ score range ^c^	−9.12 to −0.97	−0.97 to 0.41	0.41 to 1.29	1.29 to 2.19	2.19 to 6.93	
Age (years)	52.1 (7.97) ^d^	52.2 (8.14)	52.6 (8.22)	53.2 (8.45)	54.5 (8.65)	<0.0001
BMI (Kg/m^2^) ^e^	23.9 (2.77)	23.8 (2.78)	23.8 (2.76)	23.7 (2.77)	23.6 (2.86)	<0.0001
Sex						
Male (n = 53,304)	10,261 (32.6) ^f^	10,678 (33.9)	10,777 (34.2)	10,545 (33.5)	11,043 (35.0)	<0.0001
Female (n = 104,508)	21,281 (67.4)	20,887 (66.1)	20,816 (65.8)	21,010 (66.5)	20,514 (65.0)	
Education level						
Elementary school	3583 (11.5)	4066 (13.1)	4886 (15.7)	5895 (19.0)	7992 (25.7)	<0.0001
Middle ~ high school	18,511 (59.5)	18,259 (58.6)	18,117 (58.1)	17,971 (57.8)	17,360 (55.9)	
College~	9014 (29.0)	8812 (28.3)	8205 (26.2)	7243 (23.2)	5712 (18.4)	
Alcohol consumption						
Never	15,609 (49.7)	15,511 (49.3)	15,811 (50.1)	16,196 (51.4)	16,646 (53.0)	<0.0001
Past	1258 (4.0)	1148 (3.7)	1117 (3.6)	1273 (4.1)	1294 (4.1)	
Current	14,525 (46.3)	14,780 (47.0)	14,573 (46.3)	13,986 (44.5)	13,495 (42.9)	
Physical activity ^g^						
Irregular	12,234 (38.9)	13,719 (43.6)	14,607 (46.4)	15,688 (49.9)	17,944 (57.1)	<0.0001
Regular	19,212 (61.1)	17,758 (56.4)	16,911 (53.6)	15,759 (50.1)	13,494 (42.9)	
Income (million (s) Korean won)						
Less than 1	2017 (7.8)	2206 (8.3)	2691 (10.1)	3467 (13.0)	4669 (17.3)	
1~less than 2	4636 (18.0)	4924 (18.5)	5290 (19.9)	5707 (21.5)	6552 (24.2)	<0.0001
2~less than 3	5929 (23.0)	6343 (23.9)	6079 (22.9)	5901 (22.2)	5794 (21.4)	
More than 3	13,243 (51.2)	13,088 (49.3)	12,545 (47.1)	11,522 (43.3)	10,056 (37.1)	
Marital status						
Married	28,269 (90.3)	28,224 (90.0)	27,972 (89.1)	27,270 (87.0)	26,543 (84.8)	<0.0001
Unmarried/divorced	3055 (9.7)	3142 (10.0)	3453 (10.9)	4101 (13.0)	4788 (15.2)	
Smoking						
Never	23,450 (74.8)	23,161 (73.7)	23,178 (73.6)	23,255 (74.0)	22,278 (70.9)	<0.0001
Past	4312 (13.7)	4668 (14.9)	4674 (14.9)	4508 (14.3)	4562 (14.5)	
Current	3610 (11.5)	3605 (11.4)	3632 (11.5)	3680 (11.7)	4592 (14.6)	
Menopause status						
Post-menopause	10,986 (55.9)	11,053 (55.9)	11,360 (57.4)	12,050 (60.3)	12,994 (65.2)	<0.0001
Pre-/peri-menopause	8644 (44.1)	8731 (44.1)	8427 (42.6)	7943 (39.7)	6931 (34.8)	
Family history of diabetes						
Negative	25,796 (81.8)	25,901 (82.1)	25,834 (81.8)	26,010 (82.5)	26,451 (83.9)	<0.0001
Positive	5746 (18.2)	5664 (17.9)	5759 (18.2)	5545 (17.5)	5106 (16.1)	
Family history of hypertension						
Negative	22,605 (71.7)	22,188 (70.3)	22,293 (70.6)	22,587 (71.6)	22,986 (72.9)	<0.0001
Positive	8937 (28.3)	9377 (29.7)	9300 (29.4)	8968 (28.4)	8571 (27.1)	
Waist circumference (cm)	80.5 (8.4)	80.4 (8.4)	80.4 (8.3)	80.2 (8.3)	80.4 (8.4)	0.05
Triglycerides (mg/dL)	117.2 (78.3)	118.6 (78)	118.6 (79.2)	119 (80)	121.2 (82.5)	<0.0001
HDL-C (mg/dL)	55.3 (12.8)	55 (12.7)	54.8 (12.7)	54.7 (12.7)	54.3 (13)	<0.0001
Glucose concentration (mg/dL)	93.5 (20)	93.4 (18.4)	93.8 (19.5)	93.8 (19.8)	94 (20.2)	<0.0001
Blood pressure (mmHg)						
Systolic	121.5 (15)	121.4 (14.9)	121.7 (15)	121.8 (15.1)	122.2 (15.3)	<0.0001
Diastolic	74.3 (9.9)	74.3 (9.7)	74.4 (9.8)	74.4 (9.7)	74.6 (9.8)	0.05

^a^ Dietary inflammatory index (DII^®^) is presented by quintile (Q1 to Q5) at baseline, Q1 shows the anti-inflammatory index of DII^®^, while Q5 shows the maximum pro-inflammatory index of food parameters. ^b^ Jonckheere–Terpstra and Mantel–Haenszel Chi-square test was used to calculate *p* values for trend for continuous and categorical variables, respectively. ^c^ DII^®^ score range is calculated by dividing DII^®^ score into quintiles with same number of sample size (Q1~Q5) in the control group, subsequently confirmed the boundary value (minimum or maximum value) of the divided quintiles score and set the DII^®^ value of the metabolic syndrome (MS) cases. ^d^ The data for continuous variables are presented as means (standard deviation). ^e^ BMI (body mass index; Kg/m^2^) was estimated according to Asia-Pacific guidelines. ^f^ The data for categorical variables were presented as n (%) among all the participants. ^g^ Regularity of physical activity was measured according to whether or not participants contributed regularly in any sports to the point of sweating. HDL-C: high-density lipid-cholesterol.

**Table 2 nutrients-12-01196-t002:** Hazard ratios (HRs) and 95% confidence intervals (CIs) for risk metabolic syndrome and its components according to DII^®^ quintiles.

		Quintiles of DII^®^ Score ^a,b,c^					*P* for Trend ^d^	DII^®^ Continuous
		Q1	Q2	Q3	Q4	Q5		
Cases/Persons-years		618/245,414	704/236,148	732/231,182	694/231,190	696/220,584		
DII^®^, median (IQR)								
All		−2.05 (1.46)	−0.20 (0.68)	0.91 (0.43)	1.64 (0.40)	2.86 (0.96)		
Men		−2.05 (1.47)	−0.20 (0.69)	0.91 (0.43)	1.64 (0.39)	2.95 (1.06)		
Women		−2.05 (1.45)	−0.20 (0.68)	0.90 (0.43)	1.65 (0.40)	2.82 (0.89)		
Metabolic syndrome								
All		1.00 (ref.)	**1.16 (1.03−1.29) ^c^**	**1.25 (1.12−1.41)**	**1.26 (1.11−1.43)**	**1.31 (1.15−1.49)**	0.002	**1.02 (1.01−1.04)**
Men		1.00 (ref.)	1.04 (0.88−1.24)	1.09 (0.91−1.31)	**1.23 (1.01−1.48)**	1.14 (0.93−1.39)	0.39	0.98 (0.96−1.01)
Women		1.00 (ref.)	**1.23 (1.06−1.42)**	**1.36 (1.17−1.58)**	**1.28 (1.08−1.50)**	**1.43 (1.21−1.69)**	<0.0001	**1.04 (1.02−1.07)**
Elevated waist circumference								
All		1.00 (ref.)	**1.13 (1.03−1.22)**	**1.18 (1.08−1.29)**	**1.23 (1.11−1.35)**	**1.37 (1.24−1.51)**	<0.0001	**1.06 (1.04−1.07)**
Men (≥90cm)		1.00 (ref.)	1.15 (0.99−1.33)	**1.25 (1.07−1.47)**	**1.20 (1.02−1.42)**	**1.28 (1.08−1.52)**	0.01	**1.06 (1.03−1.09)**
Women (or ≥85cm)		1.00 (ref.)	**1.12 (1.01−1.24)**	**1.14 (1.02−1.27)**	**1.25 (1.11−1.40)**	**1.42 (1.26−1.61)**	<0.0001	**1.06 (1.03−1.08)**
High triacylglycerol (≥150 mg/dL),								
All		1.00 (ref.)	**1.15 (1.07−1.23)**	**1.18 (1.11−1.27)**	**1.20 (1.11−1.30)**	**1.24 (1.14−1.35)**	<0.0001	**1.03 (1.01−1.04)**
Men		1.00 (ref.)	1.10 (0.97−1.24)	1.11 (0.96−1.25)	1.11 (0.96−1.27)	1.11 (0.96−1.28)	0.23	1.01 (0.98−1.03)
Women		1.00 (ref.)	**1.16 (1.07−1.26)**	**1.22 (1.11−1.33)**	**1.25 (1.13−1.37)**	**1.30 (1.18−1.44)**	<0.0001	**1.03 (1.02−1.05)**
Low HDL-C								
All		1.00 (ref.)	**1.20 (1.08−1.33)**	**1.22 (1.09−1.37)**	**1.37 (1.21−1.54)**	**1.63 (1.44−1.84)**	<0.0001	**1.08 (1.06−1.11)**
Men (<40 mg/dL)		1.00 (ref.)	**1.26 (1.03−1.53)**	1.15 (0.93−1.42)	**1.34 (1.08−1.67)**	**1.59 (1.27−1.99)**	0.0001	**1.06 (1.02−1.11)**
Women (<50 mg/dL)		1.00 (ref.)	**1.17 (1.03−1.33)**	**1.25 (1.09−1.43)**	**1.37 (1.19−1.58)**	**1.64 (1.41−1.90)**	<0.0001	**1.09 (1.06−1.12)**
High glucose (≥100 mg/dL)								
All		1.00 (ref.)	**1.09 (1.03−1.16)**	**1.14 (1.07−1.21)**	**1.16 (1.08−1.24)**	**1.18 (1.09−1.26)**	<0.0001	**1.02 (1.01−1.03)**
Men		1.00 (ref.)	1.08 (0.98−1.18)	1.08 (0.98−1.20)	1.08 (0.97−1.20)	1.01 (0.90−1.13)	0.94	0.99 (0.97−1.01)
Women		1.00 (ref.)	**1.10 (1.02−1.19)**	**1.17 (1.07−1.27)**	**1.21 (1.11−1.32)**	**1.30 (1.18−1.43)**	<0.0001	**1.04 (1.02−1.05)**
High blood pressure								
All		1.00 (ref.)	**1.11 (1.04−1.18)**	**1.14 (1.06−1.22)**	**1.14 (1.06−1.23)**	**1.24 (1.15−1.34)**	<0.0001	**1.03 (1.02−1.04)**
Men		1.00 (ref.)	**1.16 (1.04−1.30)**	**1.13 (1.01−1.27)**	**1.16 (1.03−1.31)**	**1.17 (1.03−1.32)**	0.05	**1.02 (1.00−1.04)**
Women		1.00 (ref.)	1.07 (0.99−1.16)	**1.14 (1.05−1.24)**	**1.12 (1.02−1.23)**	**1.29 (1.17−1.41)**	<0.0001	**1.04 (1.02−1.05)**

^a^ Dietary inflammatory index^®^ (DII^®^) score presented by quintile at baseline, which divided the DII^®^ scores into five levels (Q1 to Q5). ^b^ Data are presented as hazard ratios (HRs) with correspondent 95% confidence intervals (CI). ^c^ Multivariate-adjusted for sex, age, smoke, alcohol drinking, physical activity, BMI, family history of diabetes mellitus, family history of hypertension and energy intake. *p* for heterogeneity between men and women using a likelihood test = 0.99. ^d^
*p* for trend values were determined using categorical DII^®^ scores. Metabolic syndrome was diagnosed based on the Modified National Cholesterol Education Program Adult Treatment Panel III and the obesity guidelines of the Obesity Society of Korea, as ≥3 of any of the following [27,28]: waist circumference [WC] ≥90 for men or ≥85 for women); high triglyceride level (≥150 mg/dL), low HDL-C level (<40 mg/dL in men or <50 mg/dL in women); high glucose level (fasting plasma glucose level ≥100 mg/dL) and high blood pressure (systolic blood pressure/diastolic blood pressure ≥130/85 mmgHg or the use of antihypertensive drugs). Bold letters represents significance difference at *p* < 0.05.

**Table 3 nutrients-12-01196-t003:** Hazard ratios (HRs) and 95% confidence intervals (CIs) for risk of metabolic syndrome and its components as stratified by menopausal status of women according to DII^®^ quintiles.

MS with Components			Quintiles of Dietary Inflammatory Index^®^ (DII^®^) ^a,b,c^					
	Menopausal Status	Q1	Q2	Q3	Q4	Q5	*P* for trend ^d^	DII^®^ Continuous
Metabolic syndrome	Pre-/peri-	1.00 (ref.)	1.02 (0.77−1.36)	1.02 (0.75−1.39)	1.19 (0.87−1.63)	1.26 (0.90−1.77)	0.11	1.03 (0.97−1.10)
	Post-	1.00 (ref.)	**1.27 (1.07−1.51)**	**1.47 (1.23−1.76)**	**1.28 (1.05−1.56)**	**1.50 (1.23−1.83)**	0.0008	**1.05 (1.02−1.09)**
Elevated waist circumference	Pre-/peri-	1.00 (ref.)	1.12 (0.95−1.34)	0.93 (0.76−1.13)	1.19 (0.97−1.46)	**1.42 (1.14−1.76)**	0.003	**1.06 (1.02−1.10)**
	Post-	1.00 (ref.)	**1.12 (0.99−1.28)**	**1.22 (1.07−1.40)**	**1.27 (1.10−1.46)**	**1.40 (1.21−1.63)**	<0.0001	**1.05 (1.02−1.08)**
High triacylglycerol	Pre-/peri-	1.00 (ref.)	1.12 (0.97−1.29)	1.11 (0.95−1.29)	1.15 (0.98−1.36)	**1.25 (1.04−1.48)**	0.02	**1.03 (1.00−1.06)**
	Post-	1.00 (ref.)	**1.19 (1.07−1.33)**	**1.29 (1.15−1.44)**	**1.32 (1.18−1.49)**	**1.37 (1.21−1.55)**	<0.0001	**1.04 (1.02−1.06)**
Low HDL-C	Pre-/peri-	1.00 (ref.)	1.15 (0.93−1.43)	**1.27 (1.01−1.58)**	**1.46 (1.15−1.85)**	**1.81 (1.41−2.33)**	<0.0001	**1.10 (1.05−1.16)**
	Post-	1.00 (ref.)	1.11 (0.94−1.31)	1.15 (0.97−1.37)	**1.26 (1.05−1.51)**	**1.45 (1.20−1.74)**	<0.0001	**1.07 (1.03−1.10)**
High glucose	Pre-/peri-	1.00 (ref.)	1.06 (0.92−1.21)	1.13 (0.97−1.30)	1.13 (0.97−1.32)	**1.27 (1.07−1.49)**	0.004	**1.04 (1.01−1.07)**
	Post-	1.00 (ref.)	**1.13 (1.02−1.25)**	**1.20 (1.07−1.33)**	**1.27 (1.14−1.43)**	**1.33 (1.19−1.50)**	<0.0001	**1.04 (1.02−1.06)**
High blood pressure	Pre-/peri-	1.00 (ref.)	1.08 (0.94−1.25)	1.15 (0.99−1.33)	1.12 (0.95−1.32)	**1.42 (1.20−1.68)**	0.0001	**1.06 (1.03−1.09)**
	Post-	1.00 (ref.)	1.09 (0.98−1.21)	**1.15 (1.04−1.29)**	**1.14 (1.01−1.28)**	**1.25 (1.11−1.41)**	0.0005	**1.03 (1.01−1.05)**

^a^ Dietary inflammatory index (DII^®^) score presented by quintile at baseline, which divided the DII^®^ scores into five levels (Q1 to Q5). ^b^ Data are presented as hazard ratios (HRs) with correspondent 95% confidence intervals (CI). ^c^ Multivariate-adjusted for sex, age, smoke, alcohol drinking, physical activity, BMI, family history of diabetes mellitus, family history of hypertension and energy intake. ^d^
*p* for trend values were determined using categorical DII^®^ scores. *p* for heterogeneity between Pre−/peri and post-menopause in MS, WC, TG, HDL-C, Glu and BP using a likelihood test was 0.039, <0.001, <0.001, 0.133, <0.001 and <0.001 respectively. Metabolic syndrome was diagnosed based on the Modified National Cholesterol Education Program Adult Treatment Panel III and the obesity guidelines of the Obesity Society of Korea, as ≥3 of any of the following [27,28]: waist circumference [WC] ≥90 for men or ≥85 for women); high triglyceride level (≥150 mg/dL), low HDL-C level (<40 mg/dL in men or <50 mg/dL in women); high glucose level (fasting plasma glucose level ≥100 mg/dL) and high blood pressure (systolic blood pressure/diastolic blood pressure ≥130/85 mmgHg or the use of antihypertensive drugs). Bold letters represents significance difference at *p* < 0.05.

**Table 4 nutrients-12-01196-t004:** Hazard ratios (HRs) and 95% confidence intervals (CIs) for risk of components of metabolic syndrome as stratified by smoking.

Sex	Smoking ^e^ Status	Quintiles of Dietary Inflammatory Index^®^ (DII^®^) ^a,b,c^						
		Q1	Q2	Q3	Q4	Q5	*P* for Trend ^d^	DII^®^ Continuous
**Elevated waist circumference**								
Men	No	1.00 (ref.)	1.04 (0.79−1.38)	1.07 (0.79−1.44)	1.01 (0.74−1.39)	1.18 (0.85−1.64)	0.39	1.03 (0.97−1.09)
	Yes	1.00 (ref.)	1.17 (0.98−1.40)	**1.31 (1.08−1.57)**	**1.24 (1.02−1.51)**	**1.25 (1.02−1.53)**	0.05	**1.06 (1.02−1.10)**
Women	No	1.00 (ref.)	1.10 (0.99−1.22)	1.11 (0.99−1.24)	**1.23 (1.09−1.38)**	**1.41 (1.25−1.59)**	<0.0001	**1.05 (1.03−1.08)**
	Yes	1.00 (ref.)	**2.20 (1.13−4.28)**	**3.01 (1.49−6.05)**	**2.31 (1.05−5.06)**	**2.30 (1.04−5.06)**	0.14	1.10 (0.97−1.24)
**High triacylglycerol**								
Men	No	1.00 (ref.)	**1.30 (1.01−1.67)**	1.29 (0.99−1.69)	1.23 (0.92−1.63)	1.25 (0.92−1.69)	0.33	1.02 (0.97−1.07)
	Yes	1.00 (ref.)	1.04 (0.90−1.19)	1.04 (0.89−1.21)	1.07 (0.91−1.26)	1.07 (0.90−1.26)	0.40	1.00 (0.98−1.03)
Women	No	1.00 (ref.)	**1.18 (1.08−1.28)**	**1.22 (1.11−1.33)**	**1.25 (1.14−1.38)**	**1.31 (1.18−1.45)**	<0.0001	**1.03 (1.02−1.05)**
	Yes	1.00 (ref.)	0.69 (0.41−1.16)	1.11 (0.69−1.80)	1.06 (0.63−1.78)	1.16 (0.68−1.98)	0.23	1.05 (0.96−1.15)
**Low HDL-C**								
Men	No	1.00 (ref.)	1.13 (0.78−1.63)	0.79 (0.52−1.19)	1.03 (0.67−1.58)	1.08 (0.69−1.70)	0.90	0.99 (0.92−1.07)
	Yes	1.00 (ref.)	**1.30 (1.03−1.65)**	**1.30 (1.02−1.67)**	**1.48 (1.14−1.91)**	**1.83 (1.41−2.38)**	<0.0001	**1.09 (1.05−1.14)**
Women	No	1.00 (ref.)	**1.18 (1.04−1.34)**	**1.27 (1.11−1.46)**	**1.37 (1.18−1.58)**	**1.62 (1.39−1.88)**	<0.0001	**1.09 (1.06−1.12)**
	Yes	1.00 (ref.)	0.86 (0.42−1.79)	0.84 (0.37−1.89)	1.47 (0.67−3.22)	**2.66 (1.20−5.90)**	0.005	**1.20 (1.05−1.36)**
**High glucose**								
Men	No	1.00 (ref.)	**1.20 (1.00−1.44)**	1.18 (0.97−1.42)	1.17 (0.95−1.44)	1.20 (0.96−1.49)	0.21	1.01 (0.97−1.05)
	Yes	1.00 (ref.)	1.03 (0.92−1.15)	1.05 (0.93−1.19)	1.04 (0.92−1.18)	0.94 (0.83−1.08)	0.43	0.99 (0.96−1.01)
Women	No	1.00 (ref.)	**1.11 (1.03−1.21)**	**1.19 (1.09−1.29)**	**1.22 (1.11−1.34)**	**1.29 (1.18−1.42)**	<0.0001	**1.04 (1.02−1.06)**
	Yes	1.00 (ref.)	0.81 (0.51−1.28)	0.68 (0.41−1.14)	0.98 (0.60−1.62)	1.28 (0.79−2.08)	0.10	1.03 (0.95−1.12)
**High blood pressure**								
Men	No	1.00 (ref.)	**1.30 (1.06−1.59)**	1.19 (0.96−1.48)	1.18 (0.94−1.49)	1.27 (0.99−1.61)	0.23	1.03 (0.98−1.07)
	Yes	1.00 (ref.)	1.11 (0.98−1.27)	1.11 (0.97−1.27)	**1.15 (1.00−1.33)**	1.12 (0.96−1.29)	0.17	1.01 (0.99−1.04)
Women	No	1.00 (ref.)	1.07 (0.99−1.17)	**1.13 (1.04−1.23)**	**1.11 (1.01−1.22)**	**1.26 (1.15−1.39)**	<0.0001	**1.03 (1.01−1.05)**
	Yes	1.00 (ref.)	0.91 (0.45−1.83)	1.82 (0.97−3.40)	**1.96 (1.01−3.82)**	**2.71 (1.41−5.18)**	0.0002	**1.18 (1.06−1.30)**

^a^ Dietary inflammatory index (DII^®^) score presented by quintile at baseline, which divided the DII^®^ scores into five levels (Q1 to Q5). ^b^ Data are presented as hazard ratios (HRs) with correspondent 95% confidence intervals (CI). ^c^ Multivariate-adjusted for age, smoke, alcohol drinking, physical activity, BMI, family history of diabetes mellitus, family history of hypertension and energy intake. ^d^
*p* for trend values were determined using categorical DII^®^ scores. *p* for heterogeneity between smoking and non−smoking for men and women participants using a likelihood test was <0.001, and <0.001 respectively. ^e^ No: never smoking, and yes: past/current smoking. Bold letters represents significance difference at *p* < 0.05.

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
