# Peer review of "Proinflammatory Dietary Intake is Associated with Increased Risk of Metabolic Syndrome and Its Components: Results from the Population-Based Prospective Study"

_nutrients, 2020, doi:10.3390/nu12041196_

Round 1

Reviewer 1 Report

Dear authors,

Nice work!

Some few ideas which came into my mind:

line 145-146:

I do understand what you want to tell but now it seems that the age of the paticipants would be the dependent variable instead of the DII (age depending on DII). It might be more accurate if you turn the sentence over into for example: "The DII score increased as the mean age of the participants increased." The same applies to the following lines.

Table 3:

Do you have an assumption or even explanation why non-smoking men have a higher risk of higher trigrylcerides, glucose and blood pressure in the very low quintile (Q2, i.e. less pro-inflammatory diet) in comparison to the lowest quintile (Q1), while the higher quintiles (Q3-5, i.e. more pro-inflammatory diet) do not show a higher risk compared to the reference group (Q1)?

line 245: "increased" instead of "increases"

line 275-276: "The inflammatory potential of the diet, as shown by higher DII scores, is prospectively associated with higher risk of metabolic syndrome in women" should be added, because this is not seen in men regarding these data.

Author Response

line 145-146:

I do understand what you want to tell but now it seems that the age of the paticipants would be the dependent variable instead of the DII (age depending on DII). It might be more accurate if you turn the sentence over into for example: "The DII score increased as the mean age of the participants increased." The same applies to the following lines.

Response: We have modified the sentence and made the recommended changes. (Lines 160~164)

Table 3: Do you have an assumption or even explanation why non-smoking men have a higher risk of higher trigrylcerides, glucose and blood pressure in the very low quintile (Q2, i.e. less pro-inflammatory diet) in comparison to the lowest quintile (Q1), while the higher quintiles (Q3-5, i.e. more pro-inflammatory diet) do not show a higher risk compared to the reference group (Q1)?

Response: The variable, DII is originally a continuous variable, and we categorized  the DII into 5 groups (quintiles). Cox proportional hazard analysis was performed to observe the association between DII and MetS risk. Compared to the lowest quintile (Q1), hazard ratios (HRs) was calculated for the 2nd to the 5th quintiles (Q2-Q5), respectively. As seen in other large epidemiologic studies, the HRs across successive quintiles did not increase linearly in this analysis. Accordingly, we have performed the linearity test for the DII as a continuous variable and for the 5 categories (quintiles) of DII score. We have found that p for trend using DII as a categorical variable was not significant, and p for continuous variable DII was also not significant, even though the result of second quartile (Q2) was significant. So our interpretation of this result that DII was not associated with the risk of higher trigrylcerides, glucose and blood pressure in non-smoking men, respectively.

line 245: "increased" instead of "increases"

Response: We have modified the sentence accordingly. (Line 267)

line 275-276: "The inflammatory potential of the diet, as shown by higher DII scores, is prospectively associated with higher risk of metabolic syndrome in women" should be added, because this is not seen in men regarding these data.

Response: We have modified the sentence according to the recommendation. (Line 300)

Reviewer 2 Report

The study by Khan et alia aimed at understanding the relationship between the occurrence of metabolic syndrome (MetS) and the potential pro-inflammatory properties of the diet, in a very large cohort of Koreans, using a prospective study. The authors found a positive association between consumption of a pro-inflammatory diet and risk of MetS, especially in women. The results found by Khan et alia are of great interest in the field, by linking dietary intake to long-term health consequences. While the study is well designed, well written and easy to read, the manuscript needs a few improvements, some of which are paramount for the understanding of the methodology as well as replicability and reuse. These are detailed below. In my opinion, the discussion section is well thought and well written, while the conclusions are clearly supported by the results provided herein.

I - Major concerns.

1- Throughout the manuscript, the diet inflammatory index, referred by the authors as the DII, seems to be registered (®). This fact is also briefly described on page 14, lines 286-289 in the conflict of interest section. In order for the scientific community to be able to replicate such a methodology, concerns arise with such a copyrighted index. Therefore, I assume that patients, practitioners and scientists could not freely use such a methodology in the future. Besides, the study cited in the manuscript as referring to the original methodology for calculating such a DII does not present it as “copyrighted” (study cited as number [10], by Shivappa et al, Public Health Nutr., 2014). Can the authors please clarify/explain these points? Should this index not be freely accessible, the authors should, in my opinion, explicitly state such a limitation in the method section. Besides, since my understanding of the DII, based on the explanations given in the present study as well as in other studies, is that it is calculated using mathematical formulas, how can it be registered/copyrighted? Again, could the authors please clarify/explain these points?

2- In Tables 2, 3 and 4, some statistics are written using bold numbers. Can the authors explicitly state to what they refer to? An assumption would be significantly different values?

3- The study flow chart (Figure 1, page 3) is missing some details. Can the authors explicitly give n values for every and each exclusion criteria? For example, they state that 4278 participants were excluded in the initial screening according to their energy intake, but did not provide a detailed count in both low- (n=) and high-intake (n=) men, as well as in both low- (n=) and high-intake (n=) women. Such an information could be useful for future readers to estimate prevalence of anorexia, binge eating, etc… in men and females. Please give n values for each exclusion criterion throughout Figure 1.

4- After reading the study cited as study number [10] (by Shivappa et al, Public Health Nutr., 2014), readers might be confused. Indeed, the manuscript by Shivappa et al subsequently cite the study by Cavicchia (J Nutr., 2009), where the authors mentioned that “ […] increasing Inflammatory Index score (representing movement toward an antiinflammatory diet) […] ”, while the present study by Khan et alia presents increasing DII scores as reflecting a trend towards pro-inflammatory diets. Could this induce confusion amongst readers? The authors of the present study should clarify the differences across these studies.

II - Minor points.

1- Authors forgot to give units for waist circumference (line 90, page 2). Later on, page 8 (Table 1), waist circumference is mentioned using cm, which should appear also in the text (line 90, page 2).

2- The sentence on page 5, lines 182-183 is truncated (missing verb?). Please check for grammar.

3- Two sentences are very similar. These sentences appear lines 147-150 (page 4) and contain repetitive information about drinking parameters. Please check for repetition.

4- A full stop is introduced after the beginning of the sentence on line 171 (page 5) after “ Table 3 “. Please check such a sentence for typographical error.

5- Please check the sentence written on lines 212-215 (page 13) for grammatical errors. The sentence is difficult to read, as it misses a coma.

6- In the discussion section, the authors elegantly describe the different results observed between the present study and previously-published studies. One missing element, in my opinion, is the genetic differences that could occur when studies are performed in different countries (French, Iranian, Koreans, Norwegians, etc…). Could the authors briefly mention genetic variation and their potential impact on inflammatory markers? As an example, please see Ordovas, Nutr Rev, 2007 (PMID 18240549).

Author Response

I - Major concerns.

1- Throughout the manuscript, the diet inflammatory index, referred by the authors as the DII, seems to be registered (®). This fact is also briefly described on page 14, lines 286-289 in the conflict of interest section. In order for the scientific community to be able to replicate such a methodology, concerns arise with such a copyrighted index. Therefore, I assume that patients, practitioners and scientists could not freely use such a methodology in the future. Besides, the study cited in the manuscript as referring to the original methodology for calculating such a DII does not present it as “copyrighted” (study cited as number [10], by Shivappa et al, Public Health Nutr., 2014). Can the authors please clarify/explain these points? Should this index not be freely accessible, the authors should, in my opinion, explicitly state such a limitation in the method section. Besides, since my understanding of the DII, based on the explanations given in the present study as well as in other studies, is that it is calculated using mathematical formulas, how can it be registered/copyrighted? Again, could the authors please clarify/explain these points?

Response: The Dietary Inflammatory Index (DII®) does indeed have federally registered trademark protection.   The listing of food parameters was disclosed in reference #10.  That article, published in 2014, preceded award of the federally registered trademark, which was granted in 2017.  Hence, there was no way or need to state federal registration at that time. 

Federal registration is granted based on the inherent intellectual property represented in the algorithm.  In this sense, it is no different from other trademarked algorithms (e.g., for statistical procedures such as those in SAS®, which was used for the analyses of these data). The reviewer is correct in his/her implicit assumption that a federally registered trademark is difficult to obtain.  What we provide with the DII is analogous to the nutrient calculation software that we, and many other researchers, license from the Nutrition Coordinating Center at the University of Minnesota, which provides the NDSR nutrient calculation software. An enormous amount of intellectual effort was put into developing the NDSR database and calculation software. Now, one could go to a number of different sources (e.g., USDA and food production companies) and figure this out on his or her own. However, that would be extraordinarily inefficient. It is very rare that anyone publishes a paper in this field without using some kind of nutrient database and calculation software provided commercially. We have never heard anyone demanding that NDSR divulge how they go about maintaining and organizing their database or, more importantly, actually doing the calculations. Likewise, we have never heard anyone demanding that SAS reveal the details of their procedures so that they can create statistical packages on their own.

The authors have made the calculation of the DII available to all collaborating scientists,  The company referred to in the Disclosure, Connecting Health Innovations LLC (CHI), was created for the express purpose of making DII-derived products available to the groups to which the reviewer refers; i.e., patients and practitioners. This formed the basis of an SBIR grant from the US National Institute of Diabetes, Digestive and Kidney Diseases, which would not have granted that award without assurance of a strong intellectual property position.

As we discuss in our recent perspective paper (Hebert JR, Shivappa N, Wirth MD, Hussey JR, Hurley TG. Perspective: The Dietary Inflammatory Index (DII®): Lessons Learned, Improvements Made and Future Directions. Adv Nutr 2019;10(2):185-95.),  despite revealing all 45 parameters and the literature-derived weights (which reflects ~5000 person-hours of work!), most individuals who have tried calculating DII scores on their own have failed. Consequently, we make our calculation freely available to individuals. We also do this for entire studies and this includes, for example, making the DII and the E-DII a permanent part of the Women’s Health Initiative data set.

2- In Tables 2, 3 and 4, some statistics are written using bold numbers. Can the authors explicitly state to what they refer to? An assumption would be significantly different values?

Response: Yes, a bold letter represents significantly different values. We have added the information in the footnotes of tables. (Tables 2, 3, and 4)

3- The study flow chart (Figure 1, and page 3) is missing some details. Can the authors explicitly give n values for every and each exclusion criteria? For example, they state that 4278 participants were excluded in the initial screening according to their energy intake, but did not provide a detailed count in both low- (n=) and high-intake (n=) men, as well as in both low- (n=) and high-intake (n=) women. Such an information could be useful for future readers to estimate prevalence of anorexia, binge eating, etc… in men and females. Please give n values for each exclusion criterion throughout Figure 1.

Response: the Study flow chart was corrected in Figure 1.

4- After reading the study cited as study number [10] (by Shivappa et al, Public Health Nutr., 2014), readers might be confused. Indeed, the manuscript by Shivappa et al subsequently cite the study by Cavicchia (J Nutr., 2009), where the authors mentioned that “ […] increasing Inflammatory Index score (representing movement toward an antiinflammatory diet) […] ”, while the present study by Khan et alia presents increasing DII scores as reflecting a trend towards pro-inflammatory diets. Could this induce confusion amongst readers? The authors of the present study should clarify the differences across these studies.

Response: The DII score calculated in Shivappa et al, Public Health Nutr., 2014, Khan et al. Nutrients, 2020 and in the current study is different from that of Cavicchia (J Nutr., 2009). That version, published in 2009, was the first attempt to quantify the overall effect of diet on inflammatory potential.  After publishing that DII Generation 1 (Gen1) paper in 2009 we made a number of improvements in DII Gen2, which was published five years later, in 2014 (reference #10 in this paper).  We delineate and explain the rationale for making these improvements, including reverse-scoring the DII in our 2019 paper (Hebert JR, Shivappa N, Wirth MD, Hussey JR, Hurley TG. Perspective: The Dietary Inflammatory Index (DII®): Lessons Learned, Improvements Made and Future Directions. Adv Nutr 2019;10(2):185-95). (Lines 126-128, reference #33)

II - Minor points.

1- Authors forgot to give units for waist circumference (line 90, page 2). Later on, page 8 (Table 1), waist circumference is mentioned using cm, which should appear also in the text (line 90, page 2).

Response: We have modified the sentence accordingly. (Line 97)

2- The sentence on page 5, lines 182-183 is truncated (missing verb?). Please check for grammar.

Response: We have modified the sentence accordingly. (Line 200)

3- Two sentences are very similar. These sentences appear lines 147-150 (page 4) and contain repetitive information about drinking parameters. Please check for repetition.

Response: We have corrected the text accordingly. (Line 164~166)

4- A full stop is introduced after the beginning of the sentence on line 171 (page 5) after “ Table 3 “. Please check such a sentence for typographical error.

Response: We have modified the sentence accordingly. (Line 188)

5- Please check the sentence written on lines 212-215 (page 13) for grammatical errors. The sentence is difficult to read, as it misses a coma.

Response: We have modified the sentence accordingly. (Line 230~233)

6- In the discussion section, the authors elegantly describe the different results observed between the present study and previously-published studies. One missing element, in my opinion, is the genetic differences that could occur when studies are performed in different countries (French, Iranian, Koreans, Norwegians, etc…). Could the authors briefly mention genetic variation and their potential impact on inflammatory markers? As an example, please see Ordovas, Nutr Rev, 2007 (PMID 18240549).

Response: We have included the impact of one’s genetic makeup on inflammation in the discussion section. (Line 252-255). zz